# Exploring the Experiences of Runners with Visual Impairments and Sighted Guides

**DOI:** 10.3390/ijerph191912907

**Published:** 2022-10-08

**Authors:** Lindsay E. Ball, Lauren J. Lieberman, Pamela Beach, Melanie Perreault, Jason Rich

**Affiliations:** 1Department of Human Movement Sciences, Old Dominion University, Norfolk, VA 23529, USA; 2Kinesiology, Sport Studies and Physical Education, SUNY Brockport, Brockport, NY 14420, USA; 3Wegmans School of Health and Nutrition, Rochester Institute of Technology, Rochester, NY 14623, USA

**Keywords:** blind, disability, physical activity, fitness, sport

## Abstract

Running is a popular sport, and, with simple modifications, it can be accessible for individuals with visual impairments, particularly with a sighted running guide. The purpose of this study is to examine the experiences of runners with visual impairments and sighted running guides. Adopting a descriptive qualitative approach to guide data collection analysis and interpretation, seven runners with visual impairments and four sighted running guides were recruited and interviewed. The analysis identified four major themes: (1) benefits, (2) barriers, (3) advocacy, and (4) communication. The identified themes illustrate the influence of participation in running on the health and relationships of the runners with visual impairments, the barriers that exist to participation, and the advocacy and communication needed to overcome those barriers.

## 1. Exploring the Experiences of Runners with Visual Impairments and Sighted Guides

Track and field events have been part of the Paralympics since the first games in 1960; these events are the most popular sport [1]. Track and field events, as well as distance running, have been easily adapted for individuals with visual impairments as the rules are nearly identical to those of USA Track and Field, with the most significant modification being that athletes with severe visual impairments run with a human guide [2,3]. Severity of visual impairment is determined by the visual classification system, where B1 vision is defined as blind, no light perception in either eye and an inability to recognize the shape of a hand at any distance in any direction; B2 vision is considered travel vision, the ability to recognize the shape of a hand to the visual acuity of 20/600 and/or a visual field of less than 5 degrees in the best eye with the best eye correction; and B3 vision is represented as low vision, visual acuity above 20/600 to the visual acuity of 20/200 and/or a visual field of less than 20 degrees and more than 5 degrees in the best eye with the best eye correction [4]. Runners with visual impairments’ participation in races is possible by sighted runners’ willingness to serve as a guide (e.g., guide runners) during training sessions and races [5]. Guide runners should be taught how to communicate effectively with their runners before they take on this role, regardless of the level of competition [6]. 

The United States Association of Blind Athletes (USABA) was formed to provide sporting opportunities for individuals with visual impairments through regional, national, and international competitions, as well as providing resources for sports education, training, and mainstream sports participation [7]. Positive participation in physical education and extracurricular sports provide individuals with visual impairments with the tools to reach high-level competition [7,8].

Unfortunately, adults with visual impairments have higher rates of sedentary behavior than their sighted peers [9,10]. This is due, in part, to the many barriers to physical activity that individuals who are blind or visually impaired commonly encounter, including lack of access to transportation, facilities, accessible sports, needing to rely on others, and receiving unwelcome attention from sighted individuals [11]. Nevertheless, research suggests that when individuals with visual impairments engage in physical activity, they have a higher health-related quality of life [12]; thus, it is important to encourage increased physical activity participation in this population.

Ideally, individuals with visual impairments learn the fundamental motor skills necessary to live an active life and the ability to participate in sports if desired [13]. Unfortunately, even though students with visual impairments are included in the physical education classroom, they often miss out on achieving the goals of physical education [14]. Students with visual impairments have been excluded from physical activities by physical educators and experience lowered expectations in terms of their abilities and performance [15,16,17]. According to Haegele, Zhu, and colleagues (2017) [8], individuals who went on to be Paralympic athletes experienced some form of exclusion during class and frequently chose to participate in activities that were more accessible to them, such as lifting weights. Some of the athletes had accommodating physical education teachers who provided active communication and inclusive choices that led to positive outcomes during lessons [8]. These high-level athletes attributed much of their success in sports to extracurricular blind sports opportunities such as playing goalball after school and sports education camps [8]. Ponchillia and colleagues (2002) [7] found that students with visual impairments who were enrolled in secondary physical education were more likely to participate in school and college sports. Additionally, those who played school and college sports were more likely to participate in sporting events in their community that were open to the public [7]. 

To date, there are few qualitative research studies that explore the running experiences of individuals with visual impairments. Holland and colleagues (2020) [18] examined the experiences individuals with visual impairments had when learning to run during physical education classes, while Lieberman and colleagues (2019) [19] explored the perceptions of adults who are blind related to their guide dogs that were trained for running. Runner athletes with more severe visual impairments (B1 and B2) train and compete with sighted running guides in elite events as well as in local road races [3]. If the facilitators and barriers for runners and guides are identified, then physical education and sports programs can make informed changes to encourage the participation of individuals with visual impairments in running sports events. There has been no qualitative research specific to high-level runners with visual impairments and sighted running guides. Therefore, the purpose of this study is to examine the experiences of runners with visual impairments and sighted running guides.

## 2. Methods

In order to explore the running experiences of runners who are blind or visually impaired and sighted guides, this study adopted a descriptive qualitative approach. Descriptive qualitative inquiry is a research approach that is categorical and interpretive; it produces a complete and valuable end-product while being flexible in its sampling procedures, data sources, data collection, and analysis [20]. In this current study, the researchers examined the participants’ lived experiences in their journey to becoming runners and guides. The primary focus of this study is to provide an account of the runners’ and guides’ lived experiences with recreational and high-level running while also understanding their perceptions and beliefs surrounding their running experiences. 

### 2.1. Participants

Participants for this study were recruited based on the following criteria: identifying as (a) a runner with a visual impairment or (b) a sighted running guide, (c) being 18 years of age or older, and (d) willing to participate in a recorded video conference interview. For the purposes of this study, participants were considered a runner if they were running regularly and had competed in a recreational or competitive running event in the past two years. A runner was considered a sighted running guide if they recently and frequently served as a guide for a visually impaired runner. This study protocol was accepted by the Institutional Review Board at an American university (university anonymized).

Potential participants were identified based on their previous involvement in the running community, where they have interacted with one of the researchers, through internet searches, or through snowball sampling, where participants provided the contact information of individuals who met the inclusion criteria once their own interview was complete. All participants were contacted via email to gauge their interest in participating in the study. They were informed of the purpose of the study, the eligibility criteria, the time commitment, and the incentive to participate (USD 25.00 Amazon gift card) in the study. Additionally, a link was provided in the email for individuals interested in participating in the study to read and digitally sign the informed consent that outlined the procedures of the study. Once the informed consent was signed, the lead researcher scheduled an individual interview with the eligible participants. 

Seven runners who were blind or visually impaired (19–53 years old) and four guides (21-65 years old) met the inclusion criteria, provided informed consent, completed the individual interview, and were included in the study. Three participants reported having B1 vision, two participants had B2 vision, and two participants had B3 vision per the United States Association of Blind Athletes’ visual classification criteria [4]. Five runners had a visual impairment since birth, and two acquired their visual impairment during adulthood. One runner began running in 5th grade, three began running while in middle school, two were in college or law school, and one was in their thirties. Two guides began guide running while in high school, one in college, and one in their thirties. Both the runners with visual impairments and guides have run in various events such as sprints, distance events (5K to marathons), ultra-marathons, and triathlons, where they have competed against both runners with visual impairments and sighted runners. Further demographic information specific to the runners with visual impairments can be found in Table 1; information specific to the guides can be found in Table 2. Each participant was given a pseudonym to protect their identity. 

### 2.2. Data Collection

Semi-structured audio-recorded individual interviews with the participants served as the primary source of data for this study. The semi-structured interview guides for the runners with visual impairments and sighted guides were developed by the lead researchers and sent to five experts in the field to confirm face and content validity. There were two professors in motor behavior with expertise in visual impairment, one professor of sport psychology with a background in blind sports, one professional preparation student who is visually impaired and a triathlete, and one sighted guide for sports outside of running. Some of the guided interview questions for the runners included: (1) How and when did you get involved in running? (2) What do you find to be the benefits of your running experiences? (3) What do you attribute as helpful to your success in running? (4) What do you attribute as challenging to your success in running? (5) What was the process like to find a sighted guide to run with you? Some of the guided interview questions for the sighted guides included: (1) How did you get involved in running as a sighted guide? (2) What do you find to be the benefits of your guide running experiences? (3) What do you attribute as helpful to your success in guide running? (4) What do you attribute as challenging to your success in guide running? (5) Tell me about the process of getting matched with a runner with a visual impairment. Two researchers were present in each interview to ensure trustworthiness and minimize bias by the comparison of the researchers’ interview notes. The interviews were conducted via video conferencing software (Zoom) to allow individuals across the United States to participate. Each interview began with the researchers explaining the purpose of the study and describing their personal and professional history to expose their positionality [21]. Several demographic questions were asked, followed by the guided interview questions, subsequently probing more deeply through open-form questions to obtain more detail [22]. 

Reflective interview notes documented by the researchers during each interview served as a secondary data source. The researchers noted reflections about initial feelings related to participants’ responses, participants’ non-verbal behavior, responses that seemed meaningful, and thoughts about potential themes [21]. With the interview notes, researchers were able to triangulate the data from the interviews as well as identify and consider any possible biases that may have influenced the interpretation of the interviews. 

### 2.3. Data Analysis

Each interview recording was transcribed verbatim. Participants were emailed their transcripts to allow for member-checking to enhance trustworthiness [23]. All but one participant (91%) responded to the member-checking follow-up, where they confirmed the accuracy of the transcription. The interview transcripts and reflective interview notes were analyzed by two researchers. First, the researchers independently coded the data for themes, subthemes, and supporting quotes to fully understand the wholeness of the transcript data [24]. The researchers used Braun and Clark’s (2006) [25] recommendations related to completing thematic analyses to ensure their analysis was performed in a theoretical and systematic manner. Upon completing their initial coding, the two researchers met to discuss variations of codes, assess common codes, and review emerging themes. During these discussions, the themes and subthemes were developed, and continuous analysis of the data through repeated examination occurred [26]. 

Lastly, a “critical friend” read each of the transcripts as well as the themes and subthemes found to ensure that all themes were reflective of the data collected and were in alignment with what the runners and guides shared in the interviews. Since the two researchers conducting the study were runners and knew some of the participants, the unbiased perspective of an additional researcher who was outside of the running community and yet familiar with the field was tantamount [27]. This “critical friend” held the researchers accountable and helped to ensure an unbiased view. 

Once the three researchers agreed upon the findings, the major themes and subthemes were reduced in terms of similarity of meaning and content. The themes and subthemes, along with the quotes presented below as findings, were agreed upon by all researchers. 

### 2.4. Trustworthiness

In this qualitative analysis, there were several strategies used to ensure trustworthiness. First, the interviewers took reflective notes during the interviews and stayed on the Zoom call upon the completion of each interview. The overall interview and the main points that stood out for each interviewer were discussed and documented in the notes [21]. In addition, at the beginning of each interview, the researchers exposed their positionality. This explanation of their viewpoint gave the participants insights related to their professional and personal perspectives on the topic of running for people who are blind or visually impaired [28]. In addition, a “critical friend” was utilized when analyzing the themes and subthemes to acknowledge and reduce researcher bias [27]. Lastly, 91% of the participants provided feedback on their interview transcript through the member-checking process [23]. Due to these systematic measures taken to ensure accuracy, the authors believe the data gathered represent a true reflection of what the participants have experienced in their lives.

## 3. Findings

In this study, seven blind or visually impaired runners and four sighted guides shared their experiences with running. The descriptive qualitative inquiry [20] garnered data that was further analyzed using the steps of thematic analysis [25]. As a result, the following themes were constructed: (1) benefits of running, (2) barriers within running, (3) advocacy related to running experiences, and (4) communication required during running experiences. We centered our participants’ voices by presenting our findings with direct quotes from the runners and guides in this section. 

### 3.1. Benefits

Runners and guides detailed the numerous benefits of their running experiences. Of those benefits, the participants expressed the significance of both the benefits to their *health and fitness* as well as to their *friendships*, which are the two subthemes.

#### 3.1.1. Health and Fitness

First, participants spoke about how important running was for their health and fitness. Erin (runner) summed up the health and fitness benefits by saying: “So, there’s definitely the physical health benefits that like I think all runners benefit from. Same for mental health benefits.” Both Becca and Stephanie explained that running started out as a way to be active and stay in shape but quickly transformed into valuable mental health benefits. 

Becca (runner) explained: 

I find that if I don’t run for a while I kind of start getting you know my mental health takes a little bit of a dip. I don’t have any diagnosed mental health problems, but I find that I’m a little more down, I don’t sleep as well, maybe a little more restless.

Stephanie (runner) seconded this notion by saying: 

It’s turned running into, before running was just a way for me to lose baby weight after having two kids, something to keep me busy. It’s truly turned into a mental health, a way for me to get out of my own head and to find a place of joy. 

Chris (runner) highlighted the emotional benefits he gets from running by sharing his struggle with depression while losing his vision: 

I think the largest benefit is emotional. I mean it’s everything. Running is part of who I am and it’s just a part of who I am. It’s my DNA. It’s not like I run for fun, or I run because I have to, or because I run because of exercise or I’m running to something or away from something, I just run because I run that’s part of Chris. The benefits of it, I think this is true for me and for other people, but speaking just for me, a huge emotional benefit. When I finally did confront my eyesight, I went through a severe depression and that lasted about, there was a three-year time period I was working through that, and my medicine really ended up being running.” 

Guides spoke about physical fitness being beneficial for them individually and in terms of guiding ability. Susan (guide) said: 

Well one, it keeps me in shape, you know, like somedays when I’m thinking maybe I wouldn’t run today I’m like, I have to because I said I would. And then I also feel like it’s a service if I’m going to do it and I can help open the door for somebody else to run, then it’s going to be beneficial for both of us. 

Ian (guide) explained: “I think physical fitness also played a key role because I think the more fit you are, the wider range of athletes you can take and take running with you.”

#### 3.1.2. Friendships

Aside from health and fitness, the runners and guides described the greatest benefit of running as friendship. Each runner and guide detailed the bond they formed with individual runners or the sense of community they experienced in a running group. 

Rachel (runner) shared that: 

Because of running, I got a best friend. She’s still my best friend now 12 years later, which is crazy. She’s my best friend, and I think we would have been friends, but we built that really strong bond and now she’s my person, she’s my best friend and I don’t think it would have been the same without running. 

Erin (runner) explained: 

So, a lot of my guides are friends of mine in the real world. It’s a weird dynamic, I don’t love it but I also don’t hate it because when I first started running all of my guides were friends of mine because of course I’m going to spend hours tied to you running I might as well like you as a person and I think that’s important and I definitely have races like marathons it takes a special person to want to be tied to them for 3 to 4 h. 

Phoebe (guide) expressed that:

I just like the friendship. I really enjoy being Monica’s friend. I don’t really feel like I’m doing anything extra nice. I just feel like I’m going for a run with a friend. I know for some people who volunteer it probably makes them feel really good and gives that athlete the freedom to partake in a sport. But I think it’s just really Monica and I get annoyed with life and sometimes we just need to go for a run and kind of either sprint or you know, vent about our life or whatever. 

William (guide) described the benefit of friendship in his own words as:

I feel a great sense of enjoyment and accomplishment by having a blind runner achieve their goals while at the same time that comradery that I have with another individual and to be able to bond with them as a colleague and a friend. I think that I’m the type of person that if I’m going to go out and guide someone, I’m going to create that camaraderie and they’re going to create it back with me. We’re a team, we’re going to have a good time and I’m going to build a relationship with that individual as they do with me. Many of those people that I have guided we stay very much in contact either through social media or periodically we’ll just call each other and keep in touch. It’s that friendship that is very much mutual.

Stephanie (runner) spoke highly of her running group and how they have become a close-knit family: 

Running has brought a community to me. My Achilles running group is not just a group of friends we’re family. We laugh together, we cry together, it’s very therapeutic.” Becca (runner) shared how running and her running group have remained constant social supports for her: “Yeah, I joined my local chapter of Achilles through running. You know it’s been a good social outlet for me when I didn’t really have anything else I always had Achilles and I always had running.

### 3.2. Barriers

The most substantial barrier experienced by the blind runners was the challenge of finding sighted runners to serve as guides. 

Erin stated: 

Finding people to run with. I mean you can have the biggest network in the world but sometimes when push comes to shove people just have their own training or they say they want to guide and then they realize the responsibility that comes with it, and they get nervous and back out. 

Chris recalled: 

You know it’s hard for me to find guides because a lot of times the guides would be giving up their own race to guide me. Like with this race in New York I mean the people who guided me probably could have finished you know top 10 at this race, but they don’t race, and they give up their race to guide me and there’s not a lot of people who will do that. People who guide me are like, people who guide blind runners are awesome to begin with. 

Both Erin and Rachel spoke about the added coordination of finding guides and how it can be taxing to be continuously coordinating people to run with. 

Erin said: 

…but it’s exhausting to have to constantly be trying to find people to do workouts or races with. There are times where I just don’t want to have to post on social media or email a run club. You almost kind of have to pitch your case which feels kind of weird almost not dehumanizing but I feel like it gets exhausting to constantly be asking strangers to run with you and you have to meet up with them and you have to teach them how to guide which is quick but an added layer of that first run isn’t going to be great and sometimes you only run once or twice with somebody so constantly kind of having to be on guard for that. 

Rachel explained: 

It takes constant advocacy to really find somebody to run with and have that consistency, and I’ve lived in pretty rural areas, I would say so when you’re like hey I want to run a marathon people are kind of like you what?? It’s kind of daunting and it’s a big commitment, and so I’d say that’s it and it’s also just all that coordination. So, if you have a bunch of people to run with how’s about going to be set up and then I’m also pretty much I try not to push it a lot so if somebody says yes but then like doesn’t really answer then I have to find the line of do I keep bothering this person and advocate or do I just move on? 

### 3.3. Advocacy

The runners and guides spoke about the importance of advocacy when running recreationally and in races. Erin highlighted the positive impact advocacy has had on her running experiences by stating: 

I know when I was first starting off as a runner I really got to develop as a self-advocate and got really good at kind of advocating for myself in situations that I had never had to before. I think those things kind of go beyond just the general everybody benefits from running in these ways, but I think blind runners in particular benefit from running in the like social advocacy stand points. 

Erin further explained how she has incorporated advocacy throughout her running career: 

Yeah, so, I mean everything from my preferences when it comes to running like what cuing I like, what timing I like to pacing. I ask my guides to manage pacing for me since I can’t actively look at a watch. To talking to race directors especially if I’m trying to qualify for something I’ll often advocate for a slightly early start similar to what you would give wheelchair runners to avoid some crowd clusters in the beginning. And then I think just the debrief after the race of what went well, what didn’t go well, what could be better, how could we improve, which I think is a very mutual conversation that you have on both ends but involves a lot of general advocacy because obviously as the blind person you know your experience the best and you know what you need the best. It’s a good tool I mean I teach it to young athletes how to advocate. I have them talk to their guides and explain what they need even if it is very general and has to be met with follow-up.

Sarah (runner) ran track in high school, where she and her coaches had to advocate for accommodation at meets and to ensure she had a guide to run with. She explained that: 

Since it was mostly trying to get a guide to run with me maybe sometimes that was hard and sometimes it wasn’t. Also trying to teach them, sometimes it was “oh I have to leave” and then I had to quickly find somebody to retrain and kind of tell them what to do and what not to do. I think there was that and some of the self-advocating came from my coach. They were able to talk with the officials and get my guide wire and the person behind me running as an accommodation.

Runners expressed having had to advocate for themselves to enter races. Stephanie recalled a time when she received push-back from a race director: 

It was recommended to me by my guides to sign up for a sunset half marathon which fit in to my training schedule and I also needed to start training to run in the dark since the marathon was in the dark and so the race director was not very friendly and welcoming to a blind runner running her course and so we had to do a lot of advocating. She wanted me to use a waist tether, to have my guide in front of me, she was worried that we were going to take over the road and I just really pushed back. She was still really resistant. She wasn’t ok with my guides and I running but I got so much support from the community and my guides that they all shelled out the expensive race entry and it wasn’t just me who ran that race as a blind runner but there were also two other blind runners who ran the 10K and we each had a huge group of guides who paid their own race entry because I’m sure you know that at races each blind athlete gets a free guide entry for the race.

At times, guides have also provided advocacy for the blind runners they guide. Phoebe explained when she advocated for the runner she was guiding during a race: 

This particular 10K it got to a point where it was like a very thin pathway maybe two people across not even. I mean Monica and I had to get really, really, really close and an individual had pushed her particularly and I was not impressed with that, and I let him know that that was not acceptable, and he then had something to say about that. Yeah, I know, I was like “Hi, I’m clearly holding this person, we’re a guide and visually impaired” and he got really like rude and like “don’t run so slow or don’t box me out” and stuff like that.

Steve (runner) explained that it is important for him to be a support to his fellow runners with visual impairments and guides. He explained that: 

You know if I expect others to maybe be leaders in their own community, I need to be a leader in my own community. And I need to figure out like well how do you find guides all that type of stuff and to be able to share all that information. That has also been something personally that’s really driven my ambition to do more and more and more is to create a pathway for other people, not just the not just blind people, but people that would be willing to come, alongside someone with vision loss. So, it’s a personal, it’s been personal about me, it’s about being you know, setting a positive example for my children about adversity it’s about goal setting but it’s also about you know, using my passion and experiences to help others in our community.

### 3.4. Communication

Communication was discussed by both runners and guides as an important tool for success in running recreationally and competitively. Susan explained that she talks with the person she is guiding so she knows what the runner needs and wants out of the run:

Typically, you know I ask those things what terrain how far, how fast, how do you want me to guide you. How much communication Do you want because I know I was running with my friend Dan, and I’m explaining, we’re running in Surri, I’m explaining the trees I’m explaining the houses and he’s like Susan, I don’t care, I just don’t care, I don’t need to know about that crap. I was like, Okay, I don’t need to tell you about anything in the environment, but sometimes people do care. 

William further explained how he communicates with runners depending on the situation and how he communicates before and during the run: 

So, depending on the situation, if I’m out on the trail with John and we’re running and we’ve run hundreds of times up a trail and when there’s a curb, I’m going to give my voice commands “curb up, 1, 2, 3 and curb down, 1, 2, 3, biker on the left, great looking gal on the right”. And so, I am going to be that person’s eyes and I’m going to adapt to whatever the situation is. If we’re running in a race the tether is going to be a lot tighter or a lot looser as opposed to running on a trail. It really depends on the athlete, the race, and what we’re trying to accomplish that day, but my number one goal is to keep that athlete safe and if there’s an obstacle coming up, we’re going to do arm over arm and we’re going to have those discussions first before we run to know what our technique is. I’m going to listen to them to know what his or her needs are. 

Ian discussed that in order for him to become a better guide, it was important for him to be able to take feedback from the runners he was guiding. He explained: 

Being able to take criticism and feedback really well, because usually I think the first few times you would go running, we were just testing things here, seeing what worked, what didn’t. And so being able to take criticism and go back to the drawing board, I think improves the experience for myself and the athletes. So even like the week of camp, we would run a lap around the track. Do we need to fix anything? No. Okay. Let’s go again. Yes. All right. Let’s think of ways to make it better.

## 4. Discussion

Seven runners with visual impairments and four guides were interviewed to capture the essence of their running experiences. Careful consideration of the findings and careful construction of the themes (i.e., benefits, barriers, advocacy, and communication) allowed for greater illumination and exploration of the extant literature. The following discussion will highlight relevant similarities, differences, implications, and limitations to further the overall conversation. 

The theme of benefits included both health and fitness and friendships. The subtheme of health and fitness was clearly a benefit, as individuals with visual impairments have been found to have lower levels of health-related fitness [12,29]. The implementation of running in the participants’ lives ensured that they did not succumb to the sedentary lifestyles that affect many of their peers with visual impairments [11,12,30]. Prior research has indicated that individuals with visual impairments who engage in physical activity have a greater perceived health-related quality of life that encompasses physical and mental health as well as social relations [12,31]. There have been many studies that have shown that people with visual impairments experience loneliness and isolation [32,33,34]. For this reason, the sub-theme of friendships and camaraderie found in the running community is important. According to our findings, the participants’ involvement in running helped them to create life-long friendships and relationships. Both runners with visual impairments and sighted guides reported numerous physical and mental health benefits that they contributed to running, such as fitness, increased energy, enhanced mood, and positive relationships with others that may positively contribute to their perceived health-related quality of life [31]. Promoting the benefits of running may encourage individuals to get involved in the welcoming sport. 

The theme of barriers highlighted the lack of guide runners. The lack of sighted running guides may be an international phenomenon, as Alcaraz-Rodriguez and colleagues (2018) [35] found that there is a shortage of sighted runners who are willing and trained to be guides in the trail running community in Spain. This is similar to findings by Lieberman et al. (2019) [19] in their study examining the experience of running guide dog handlers. Participants chose to get a running guide dog as they continued to have issues finding guides to run with. The specific barriers experienced by participants in Lieberman and colleagues’ 2019 study [19] were the time the guides had to run, pacing, and communication. The barriers to running in our current study were similar, spanning the time it takes to find guides on a regular basis, guides wanting to run their own races rather than guide someone, and the need for constant advocacy; these barriers are exhausting for runners with visual impairments. Aligning with the barriers identified by research on elite athletes with physical disabilities, these barriers can play a role in the continued motivation of runners with visual impairments [36].

Although the constant advocacy was tiresome for runners, it was clearly linked to individual and community success. The need to advocate was found to be critical in Lieberman et al.’s 2019 study [19] on running guide dogs. It appears that finding a guide that matches the runner with a visual impairment in terms of the time of day, pace, and goals is an issue. The current study also highlights the need for guides to advocate for the runners they guide in some cases. Being aware of the needs that runners with visual impairments often advocate for can enable race officials to prepare for and welcome runners and guides into the race community. The skill of advocacy is rarely taught to youth in schools, and this has come up recently as a major needed area in order to be successful [37]. 

Ensuring students with visual impairments are taught the skills to advocate is critical for a higher quality of life as well as to obtain guide runners when needed. However, through their sport participation and competition, runners with visual impairments can develop a sense of empowerment in order to advocate for themselves and others [36].

The theme of communication was no surprise as communication is necessary for effective guide running [38,39]. Effective communication between runners with visual impairments and their guide runners is interdependent, and collaboration is necessary for long-term success [40]. At this time, runners with visual impairments often train new guides as needed. For example, our participants found guides to run with in their community, many of whom had little or no guiding experience. In a study by Rudiyati (2014), [6] teacher candidates who were studying how to teach youth with visual impairments went through a protocol with their students on how they could work as sighted guides. They had an evaluation system related to how they provided guidance and feedback. This type of systematic protocol for the communication of guide runners may be helpful when a runner with a visual impairment is training a new guide. 

## 5. Limitations and Future Research

This study had a relatively small sample size, although, for a qualitative study, it was sufficient [41]. In addition, there were two sources of data as well as the perspectives of runners and guides. The participants’ level of running or success was not addressed as part of the criteria.

Future research should include family members, coaches, friends, and colleagues to gain additional perspectives for triangulation. In addition, the researchers can add observations of runners and guides during practice and races. Additionally, future research can analyze the lived experiences of new and seasoned runners. Lastly, future qualitative inquiries can explore how runners with visual impairments and sighted runners become involved in the sport as an athlete or a guide.

## 6. Conclusions

Through the results of this descriptive qualitative study, it was clear that running has many benefits, which include health and fitness as well as friendship and socialization. Based on these findings, running is a good physical activity option for individuals with visual impairments to participate in. Running is an activity that allows people to maintain health and fitness while creating meaningful relationships. Although there are numerous benefits, there is a consistent barrier to finding guides, which illuminates the need for advocacy for the runners. In some cases, it also is necessary for the guides to advocate for the runners. Lastly, there is a need for clear and consistent communication between the runner and the guide. Implications for training youth with visual impairments using a protocol on guide running have been shared as well as future directions for research for this popular and enjoyable sport of running.

## Figures and Tables

**Table 1 ijerph-19-12907-t001:** 0 Demographic information of the runners.

Name	Age	Gender	VI Classification	Educ.	Demographic	Began Running	Events
Erin	24	F	B2	BS	Urban	Middle Sch.	Distance and Tri
Chris	51	M	B3	JD	Suburban	Law school	Distance and Tri
Rachel	29	F	B1	MS	Suburban	Middle Sch	Distance Marathon
Becca	24	F	B1	BS	Suburban	5th grade	Distance
Sara	19	F	B2	Associates	Rural	Middle Sch	Sprints
Steve	53	M	B1	Masters	Suburban	College	Distance Ultras
Stephanie	38	F	B2	BS	Suburban	30s	Distance Ultras

**Table 2 ijerph-19-12907-t002:** 0 Demographic information of the guides.

Name	Age	Gender	Yrs Guiding	Level of VI	Began Guiding	Events	Education Level	Demographic
Susan	56	F	33	B1–B3	College	Distance	Ph.D.	Suburban
William	65	M	28	B1–B4	30’s	Sprints Distance	Masters	Rural
Ian	21	M	3	B1–B2	HS	Sprints Distance	HS	Suburban
Phoebe	28	F	13	B1	HS	Sprints Distance	MS	Rural

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
