# Peer review of "Exploring the Experiences of Runners with Visual Impairments and Sighted Guides"

_ijerph, 2022, doi:10.3390/ijerph191912907_

Round 1
Reviewer 1 Report
Dear Authors,
The article take into consideration an important topic, i.e. activation of the blind and visually impaired people. Sport of people with disabilities is very often undertaken in research. However, most of the research is quantitative. That is why I see a great value in the qualitative approach. I can see a lot of work needed to do this research. However, it seems that they may be of interest to a narrow community. The more so because they are based on a small group of respondents, which makes generalization and inference difficult. For this reason, the article is more of a cognitive than an application form.
The introduction does not raise any objections.
The Methods and Participants are clearly described. The small group of respondents and the large age differentiation raise reservations.
Chapters 8, 9, and 10 should be a subsection for Findings. Moreover, "advovacy" should also be a subsection.
To sum up, it seems that the work is not revealing. The benefits of practicing physical activity for blind people are well known. The barriers to taking up activity are also known. The issue of the relationship between a blind runner and a guide seems to be interesting. Perhaps this is the specific direction of qualitative research in this group.
Please indicate what innovative your article brings? Where do you see his contribution to the current state of knowledge?
I would also like to point out that the authors cited 11 of their own works, which constitutes 25 percent of all references.
Author Response
Dear Reviewer,
Thank you for your comments. Below is your feedback with responses from the authors.
Comments and Suggestions for Authors
Dear Authors,
The article take into consideration an important topic, i.e. activation of the blind and visually impaired people. Sport of people with disabilities is very often undertaken in research. However, most of the research is quantitative. That is why I see a great value in the qualitative approach. I can see a lot of work needed to do this research. However, it seems that they may be of interest to a narrow community. The more so because they are based on a small group of respondents, which makes generalization and inference difficult. For this reason, the article is more of a cognitive than an application form.
Author Comment: Thank you for your feedback.
The introduction does not raise any objections.
Author Comment: Thank you.
The Methods and Participants are clearly described. The small group of respondents and the large age differentiation raise reservations.
Author Comment: Thank you for your comment. We have added a citation confirming the sample size is sufficient. The running population encompasses a wide age demographic; therefore, we believe our sample is appropriate considering this.
Chapters 8, 9, and 10 should be a subsection for Findings. Moreover, "advovacy" should also be a subsection.
Author Comment: Thank you for pointing this out. The sections and subsections have been changed.
To sum up, it seems that the work is not revealing. The benefits of practicing physical activity for blind people are well known. The barriers to taking up activity are also known. The issue of the relationship between a blind runner and a guide seems to be interesting. Perhaps this is the specific direction of qualitative research in this group.
Please indicate what innovative your article brings? Where do you see his contribution to the current state of knowledge?
Author Comment: Thank you for your comment. We believe this manuscript contributes to knowledge in the field as it explored the lived experiences of runners with visual impairments and sighted guides. The benefits and barriers were expressed by the individuals themselves. We believe that moving forward practitioners, runners, guides, coaches, teachers, and race officials can use this information to better understand the process of running for the athlete/guide team.
I would also like to point out that the authors cited 11 of their own works, which constitutes 25 percent of all references.
Author Comment: This is such a small field and as a result, the reviewer new the identity of the authors in this double-blind study. That being said, there are very few researchers in the field conducting research in this important area and therefore the use of all available resources necessitates referencing all that is out there. We hope that this study encourages more people to do research on this unique topic.
Reviewer 2 Report
Dear authors,
First of all, thank you for submitting the manuscript entitled “Exploring the Experiences of Runners with Visual Impairments and Sighted Guides” for consideration for publication in the International Journal of Environmental Research and Public Health, Sport and Health Section, Special Issue on “Health, Physical Activity, and Recreation in Individuals with Visual Impairments, Deafblindness, or Visual Impairment with Additional Disabilities”.
Authors are invited to submit papers addressing the benefits and/or the current status of health and fitness indices, motor skills, or recreation in individuals with visual impairment, deafblindness and additional disabilities across the lifespan, using longitudinal, cross-sectional, descriptive, and qualitative inquiries. Therefore, the paper is suitable for this Special Issue since it aims to examine the experiences of runners with visual impairments and sighted running guides, implementing a qualitative approach.
Since there is scarce qualitative research studies that explore the running experiences of individuals with visual impairments and sighted running guides, one of the major positive points was the contribution to this topic. Also, facilitators and barriers for runners and guides are identified.
The paper is well written and I find the manuscript interesting and very significant.
Regarding the methodology, the descriptive qualitative inquiry used is an appropriate research approach for the purpose of the study.
The participants’ lived running experiences as well as their perceptions and beliefs.
Both semi-structured audio recorded interviews and reflective interview notes by the researchers were suited for the data collection, and all the steps of data analysis and trustworthiness were accomplished.
I just have a few comments to the paper in its current form:
Page 10 – “higherquality”
Conclusion:
Regarding the “Implications for training youth with visual impairments on guide running were shared “ in my opinion the authors could highlight some of those suggestions.
Author Response
Dear Reviewer,
Thank you for your comments. Below is your feedback with responses from the authors.
Comments and Suggestions for Authors
Dear authors,
First of all, thank you for submitting the manuscript entitled “Exploring the Experiences of Runners with Visual Impairments and Sighted Guides” for consideration for publication in theInternational Journal of Environmental Research and Public Health, Sport and Health Section, Special Issue on “Health, Physical Activity, and Recreation in Individuals with Visual Impairments, Deafblindness, or Visual Impairment with Additional Disabilities”.
Authors are invited to submit papers addressing the benefits and/or the current status of health and fitness indices, motor skills, or recreation in individuals with visual impairment, deafblindness and additional disabilities across the lifespan, using longitudinal, cross-sectional, descriptive, and qualitative inquiries. Therefore, the paper is suitable for this Special Issue since it aims to examine the experiences of runners with visual impairments and sighted running guides, implementing a qualitative approach.
Since there is scarce qualitative research studies that explore the running experiences of individuals with visual impairments and sighted running guides, one of the major positive points was the contribution to this topic. Also, facilitators and barriers for runners and guides are identified.
The paper is well written and I find the manuscript interesting and very significant.
Regarding the methodology, the descriptive qualitative inquiry used is an appropriate research approach for the purpose of the study.
The participants’ lived running experiences as well as their perceptions and beliefs.
Both semi-structured audio recorded interviews and reflective interview notes by the researchers were suited for the data collection, and all the steps of data analysis and trustworthiness were accomplished.
Author Comment: Thank you for the positive feedback and acknowledgement that the manuscript is appropriate for this issue.
I just have a few comments to the paper in its current form:
Page 10 – “higherquality”
Author Comment: Thank you for bringing this to our attention. The error has been fixed.
Conclusion:
Regarding the “Implications for training youth with visual impairments on guide running were shared “ in my opinion the authors could highlight some of those suggestions.
Author Comment: Thank you for the suggestion. An example has been added in the conclusion section.
Reviewer 3 Report
INTRODUCTION
I think that sport class based on visual impairment should be described at the beginning of the introduction when talking about the Paralympics.
METHODS
There is a need to describe why it is understood by “being a runner” (for instance, I run three days per week, and I consider myself a runner).
Are blind athletes competing together with runners without visual impairment?
All participants were born with VI? This is an important fact to consider.
Were the athletes being interviewed separately?
RESULTS
I think that information related on how people become a running guide should be placed somewhere in the manuscript. This would allow for a better understanding of the results showed here.
DISCUSSION
I think that it could be useful to place some practical applications. For instance, being healthy is one of the main reasons for being a runner. How this information could help to promote an active lifestyle among people with VI (since the authors themselves mention in the introduction section the sedentary lifestyle issue)?; What can be learned by those professionals in charge of organizing running events to promote the inclusion of runners with VI?
A reference is needed to support the assumption that the sample size is accurate for a study like this.
CONCLUSION
My advise is to first show the main reasons behind wanting to be a runner with VI.
Author Response
Dear Reviewer,
Thank you for your comments. Below is your feedback with responses from the authors.
Comments and Suggestions for Authors
INTRODUCTION
I think that sport class based on visual impairment should be described at the beginning of the introduction when talking about the Paralympics.
Author Comment: Thank you for the suggestion, visual classifications have been added to the first paragraph.
METHODS
There is a need to describe why it is understood by “being a runner” (for instance, I run three days per week, and I consider myself a runner).
Author Comment: Thank you for the suggestion. We have clarified what identifying as a runner means for this study.
Are blind athletes competing together with runners without visual impairment?
Author Comment: Thank you for the question. This information has been added in the methods section.
All participants were born with VI? This is an important fact to consider.
Author Comment: Thank you for the question. We have added this information to the methods section.
Were the athletes being interviewed separately?
Author Comment: Thank you for the question. We have clarified throughout the methods section that the participants completed individual interviews.
RESULTS
I think that information related on how people become a running guide should be placed somewhere in the manuscript. This would allow for a better understanding of the results showed here.
Author Comment: Thank you for the suggestion. We believe that discussing how runners became guides is outside the scope of this manuscript. We were more focused on the lived experiences of serving as a guide. We have added this to future research. In the methods section we clarified what identifying as a sighted guides means for this study.
DISCUSSION
I think that it could be useful to place some practical applications. For instance, being healthy is one of the main reasons for being a runner. How this information could help to promote an active lifestyle among people with VI (since the authors themselves mention in the introduction section the sedentary lifestyle issue)?; What can be learned by those professionals in charge of organizing running events to promote the inclusion of runners with VI?
Author Comment: Thank you for the suggestion. We have added practical applications in the discussion.
A reference is needed to support the assumption that the sample size is accurate for a study like this.
Author Comment: Thank you. A reference has been added.
CONCLUSION
My advise is to first show the main reasons behind wanting to be a runner with VI.
Author Comment: Thank you for your advice. We have revised the conclusion.
Round 2
Reviewer 1 Report
Thank you for your anwers.